# GENERALIZING TREE MODELS FOR IMPROVING PREDICTION ACCURACY

## ABSTRACT

Can we generalize and improve the representation power of tree models? Tree models are often favored over deep neural networks due to their interpretable structures in problems where the interpretability is required, such as in the classification of feature-based data where each feature is meaningful. However, most tree models have low accuracies and easily overfit to training data. In this work, we propose Decision Transformer Network (DTN), our highly accurate and interpretable tree model based on our generalized framework of tree models, decision transformers. Decision transformers allow us to describe tree models in the context of deep learning. Our DTN is proposed based on improving the generalizable components of the decision transformer, which increases the representation power of tree models while preserving the inherent interpretability of the tree structure. Our extensive experiments on 121 feature-based datasets show that DTN outperforms the state-of-the-art tree models and even deep neural networks.

## 1  INTRODUCTION

*Can we generalize and improve the representation power of tree models?* The tree models learn a structure where the decision process is easy to follow (Breiman et al., 1984). Due to this attractive feature of tree models, various efforts are being put to improve their performance or to utilize them as subcomponents of a deep learning model (Kontschieder et al., 2015; Shen et al., 2018).

Compared to typical deep neural networks, the main characteristic of tree models is that input data are propagated through layers without a change in their representations; the internal nodes calculate the probability of an input $\mathbf{x}$ arriving at the leaf nodes. The *decision* process that determines the membership of a training example, i.e., the process of constructing the subsets of training data at each leaf, is the core operation in tree models, which we generalize and improve in this work.

Previous works on tree models consider the decisions at different nodes as separate operations, i.e., each node performs its own decision independently from the other nodes in the same layer, based on the arrival probability of an input $\mathbf{x}$ to the node. The independence of decisions allows learning to be simplified. However, in order to obtain a comprehensive view, the decisions of multiple nodes must be considered simultaneously. Furthermore, a typical tree model with a depth of $L$ requires $b^L - 1$ decision functions, where $b$ is a branching factor, which makes it intractable to construct a deep tree model.

In this work, we suggest that many tree models can be generalized to what we term as the *decision transformer* as Proposition 1. A decision transformer generalizes existing tree models by treating the decisions of each layer as a single operation involving all nodes in that layer. More specifically, the decision transformer views each layer as a *stochastic decision* (Def. 2) which linearly transforms the membership of each training data by a learned *stochastic matrix* (Def. 1). The aggregated layer-wise view allows the decision transformer to reduce the complexity of analysis to $O(L)$, where $L$ is the tree depth. Furthermore, formulating the tree models in a probabilistic context allows them to be understood by the deep learning framework and provides us a theoretical foundation for deeply understanding the advantages and the limitations of existing tree models in one framework.

The in-depth understanding of tree models as decision transformers allows us to propose Decision Transformer Network (DTN), a novel architecture that inherits the interpretability of tree models while improving their representation power. DTN is an extension of tree models into deep networks

that creates multiple paths between nodes based on stochastic decisions. We propose two versions of DTN: DTN-D with full dense connections and DTN-S with sparse connections where the edges are connected in a locality-sensitive form. Our experiments show that DTN outperforms previous tree models and deep neural networks on 121 tabular datasets.

Our contributions are summarized as follows. First, we introduce a generalized framework for tree models called the *decision transformer* and perform theoretical analysis on its generalizability and interpretability (Section 3). Second, we propose dense and sparse versions of Decision Transformer Network (DTN), our novel decision-based model that is both interpretable and accurate (Section 4). Third, we undergo extensive experiments on 121 tabular datasets to show that DTN outperforms the state-of-the-art tree models and deep neural networks (Section 5).

## 2 RELATED WORKS

A tree model refers to a parameterized function that passes an example through its tree-structured layers without changing its representations. We categorize tree models based on whether it utilizes linear decisions over raw features or neural networks in combination with representation learning.

**Tree Models with Linear Decisions**   Traditional tree models perform linear decisions over a raw feature. Given an example $\mathbf{x}$, decision trees (DT) (Breiman et al., 1984) compare a single selected element to a learned threshold at each branch to perform a decision. Soft decision trees (SDT) (Irsoy et al., 2012) generalize DTs to use all elements of $\mathbf{x}$ by a logistic classifier, making a probabilistic decision in each node. SDTs have been widely studied and applied to various problems due to its simplicity (Irsoy & Alpaydin, 2015; Frosst & Hinton, 2017; Linero & Yang, 2018; Yoo & Sael, 2019). Deep neural decision trees (Yang et al., 2018) also extend DTs to a multi-branched tree by splitting each feature element into multiple bins. These models provide a direct interpretation of predictions and decision processes. There are also well-known ensembles of such linear trees, such as random forests (Breiman, 2001) and XGBoost (Chen & Guestrin, 2016). These ensemble models tboost the performance of tree models at the expense of interpretability.

**Tree Models with Representation Learning**   There are recent works combining tree models with deep neural networks, which add the ability of representation learning to the hierarchical decisions of tree models. A popular approach is to use abstract representations learned by convolutional neural networks (CNNs) as inputs of the decision functions of tree models (Kontschieder et al., 2015; Roy & Todorovic, 2016; Shen et al., 2017; 2018; Wan et al., 2020), sometimes with multilayer perceptrons instead of CNNs (Bulò & Kontschieder, 2014). Another approach is to categorize raw examples by hierarchical decisions and then apply different classifiers to the separate clusters (Murthy et al., 2016). There are also approaches that improve the performance of deep neural networks by inserting hierarchical decisions as differentiable operations into a neural network, instead of building a complete tree model (Murdock et al., 2016; McGill & Perona, 2017; Brust & Denzler, 2019; Tanno et al., 2019). In this work, we focus on models having complete trees in its structure.

## 3 GENERALIZING TREE MODELS

We first introduce a decision transformer, our unifying framework that represents a tree model as a series of linear transformations over probability vectors. We then analyze the properties of decision transformers: the generalizability to existing tree models and the interpretability.

### 3.1 DECISIONS TRANSFORMERS

**Definition 1** (Stochastic matrix). *A matrix $\mathbf{T}$ is (left) stochastic if $\mathbf{T} \geq 0$ and $\sum_i a_{ij} = 1$ for all $j$. A vector $\mathbf{p}$ is stochastic if it satisfies the condition as a matrix of size $|\mathbf{p}| \times 1$. We denote the set of all stochastic vectors and matrices as $\mathcal{P}$, i.e., we represent that $\mathbf{T} \in \mathcal{P}$ or $\mathbf{p} \in \mathcal{P}$.*

**Definition 2** (Stochastic decision). *A function $\mathbf{f}$ is a stochastic decision if it linearly transforms an input vector $\boldsymbol{\pi}$ by a left stochastic matrix as $\mathbf{f}(\boldsymbol{\pi}) = \mathbf{T}\boldsymbol{\pi}$, where $\mathbf{T} \in \mathcal{P}$ is independent of $\boldsymbol{\pi}$.*

**Corollary 1.** *Given a stochastic decision $\mathbf{f}$ and a vector $\boldsymbol{\pi}$, $\mathbf{f}(\boldsymbol{\pi}) \in \mathcal{P}$ if $\boldsymbol{\pi} \in \mathcal{P}$.*

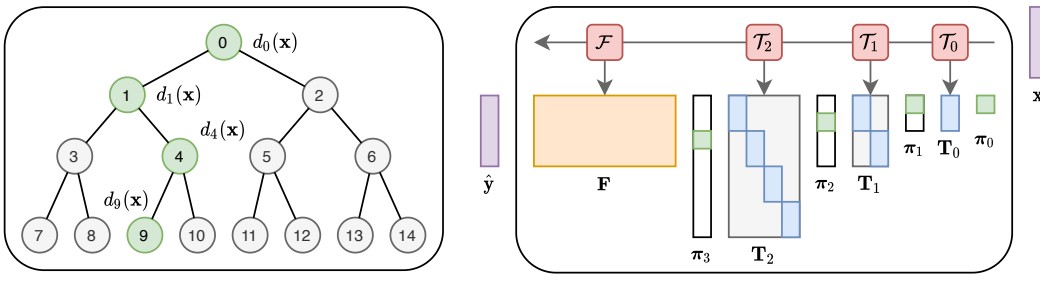

(a) A tree model by a traditional view.   (b) A tree model as a decision transformer $\hat{\mathbf{y}} = \mathcal{M}(\mathbf{x})$.

Figure 1: Illustrations of a binary tree model by (a) a traditional node-and-branch structure and (b) our representation as a decision transformer. The traditional view treats the prediction as a sequence of decisions $d_0$, $d_1$, $d_4$, and $d_9$ following the path of $\mathbf{x}$. We understand the same model as a decision transformer represented as a series of linear transformations regardless of the actual path of $\mathbf{x}$.

Based on the observation that each layer of a tree model can be represented as a stochastic decision that eventually learns the membership probability of an input $\mathbf{x}$ to the leaf nodes, we propose *decision transformers* which represent a tree model as a series of stochastic decisions whose transition matrices are determined by $\mathbf{x}$.

**Proposition 1** (Decision transformer). *A parameterized classifier $\mathcal{M}$ is a decision transformer if it can be represented as*

$$\mathcal{M}(\mathbf{x};\Theta) = \mathbf{F}\mathbf{T}_{L-1}\cdots\mathbf{T}_1\mathbf{T}_0\boldsymbol{\pi}_0 \in \mathbb{R}^C, \tag{1}$$

*where $C$ is the number of target classes, $\boldsymbol{\pi}_0 \in \mathcal{P}$ is the initial decision probability, $L$ is the number of decisions, $\mathbf{T}_l \in \mathcal{P}$ is a transition matrix, and $\mathbf{F} \in \mathcal{P}$ is a classifier matrix. $\mathbf{T}_l$ and $\mathbf{F}$ are calculated from parameterized functions $\mathcal{T}_l$ and $\mathcal{F}$, respectively, that take $\mathbf{x}$ as inputs:*

$$\mathbf{T}_l = \mathcal{T}_l(\mathbf{x};\theta_l) \in \mathbb{R}^{N_{l+1}\times N_l} \qquad\qquad \mathbf{F} = \mathcal{F}(\mathbf{x};\theta_L) \in \mathbb{R}^{C\times N_L}, \tag{2}$$

*where $N_l$ is the size of decision vectors after the $l$-th transformation. The initial decision probability $\boldsymbol{\pi}_0$ is typically given as a hyperparameter, and thus $\Theta = \{\theta_0, \cdots, \theta_L\}$.*

We define $\mathbf{T}_l$ and $\mathbf{F}$ separately since internal and leaf nodes have different functionality in most tree models, although they are basically the same. For example, the $l$th internal layer of a typical decision tree updates a one-hot membership probability of $\mathbf{x}$ by a stochastic matrix $\mathbf{T}_l$, which is generated by comparing an element of $\mathbf{x}$ to a learned threshold. In this case, $\mathbf{T}_l$ is a rectangular block-diagonal matrix having nonzero elements at $(2j, j)$ and $(2j+1, j)$ for every node $j$ in the layer. On the other hand, the leaf layer utilizes a dense classification matrix $\mathbf{F}$ that makes the final prediction.

Figure 1 illustrates two views of a binary tree model. Figure 1a treats a tree model as a set of independent decisions that are selected based on the path of $\mathbf{x}$. On the other hand, Figure 1b represents the tree as a series of linear transformations, based on our framework of decision transformers.

## 3.2 GENERALIZING BINARY TREE MODELS

We now represent existing tree models as decision transformers and determine their characteristics. We work with binary trees for simplicity, but the analysis can be naturally extended to tree models having higher branching factors (Yang et al., 2018; Murthy et al., 2016; Tanno et al., 2019). Binary tree models can be represented with block diagonal transition matrices with nonzero values on the $2 \times 1$ diagonal blocks, such that each column contains at most two nonzero elements that sum to one. Corollary 2 formalizes this notion.

**Corollary 2.** *The $l$-th transition function $\mathcal{T}_l$ of a binary tree model is represented as*

$$\mathcal{T}_l(\mathbf{x}) = \mathrm{diag}\left(\begin{bmatrix} d_1(\mathbf{x}) \\ 1-d_1(\mathbf{x}) \end{bmatrix}, \cdots, \begin{bmatrix} d_n(\mathbf{x}) \\ 1-d_n(\mathbf{x}) \end{bmatrix}\right) \in \mathbb{R}^{2^{l+1}\times 2^l}, \tag{3}$$

*where $\mathrm{diag}(\cdot)$ generates a rectangular block diagonal matrix, and $d_i$ is the unit decision function of node $i$, which measures the probability of $\mathbf{x}$ for taking the left branch at node $i$.*

Table 1: Summary of binary tree models and their representations as decision transformers: decision trees (DT), soft decision trees (SDT), neural decision forests (NDF), deep neural decision forests (DNDF), and neural regression forests (NRF). Detailed explanations are presented in Section 3.2.

| Model | $d_i(\mathbf{x})$ | $\mathcal{F}_j(\mathbf{x})$ |
|---|---|---|
| DT (Breiman et al., 1984) | $\mathbb{I}(s_i(\mathbf{1}_i^\top \mathbf{x} - b_i) > 0)$ | $\mathrm{Onehot}(\theta_j)$ |
| SDT (Irsoy et al., 2012) | $\sigma(\mathbf{w}_i^\top \mathbf{x} + b_i)$ | $\mathrm{Categorical}(\theta_j)$ |
| NDF (Bulò & Kontschieder, 2014) | $\mathrm{MLP}_i(\mathrm{rand}(\mathbf{x}))$ | $\mathrm{Categorical}(\theta_j)$ |
| DNDF (Kontschieder et al., 2015) | $\mathrm{CNN}(\mathbf{x}; i)$ | $\mathrm{Categorical}(\theta_j)$ |
| NRF (Roy & Todorovic, 2016) | $\mathrm{CNN}(\mathbf{x}; i, \mathrm{depth}(i))$ | $\mathrm{Gaussian}(\theta_j)$ |

Based on Corollary 2, we represent various tree models with specific forms of transition functions as summarized in Table 1. The unit decision function of decision trees (DT) selects a single feature of $\mathbf{x}$ by a one-hot vector $\mathbf{1}_i$ and then compares it to a threshold $b_i$ considering a learnable sign $s_i$. Each leave returns a one-hot vector of length $C$ as a prediction, where $C$ is the number of classes. Soft decision trees (SDT) extend DTs by a) producing a probabilistic decision utilizing all features of $\mathbf{x}$ at each internal node and b) returning a soft categorical distribution at each leaf as a prediction. The decision functions of these two models are linear with respect to the raw feature $\mathbf{x}$ and thus the decision processes are naturally interpretable.

Complex models that utilize deep neural networks for decision functions can also be generalized with the decision transformer. Neural decision forests (NDF) use a randomized multilayer perceptron (MLP) as a decision function. Deep neural decision forests (DNDF) use a shared convolutional neural network (CNN) for all decisions with changing only the last fully-connected layer. Neural regression forests (NRF) improve DNDFs by adopting a hierarchical CNN having different numbers of convolution operations based on the depth of node $i$ in addition to the last fully-connected layer. Other models (Shen et al., 2017; 2018) that have an identical structure to DNDFs and NRFs.

## 3.3 Advantages of Using Multiple Layers

The transition matrices of a decision transformer in Proposition 1 can be collapsed to form a single transition matrix. However, there are two practical advantages to keeping them long. First, dividing a strong black box learner into a series of weak learners improves both the generalizability and interpretability of a model. Using a deep neural network as a single strong classifier is a popular approach in many domains, but is known to overfit easily especially in tabular data having insufficient training examples. We split it into multiple weak learners which linearly accumulate their representation power by matrix multiplications. Thus, it is possible to adjust the learning capacity of each weak learner individually, based on the complexity of the target problem and avoiding overfitting.

Second, the output $\boldsymbol{\pi}_i$ from each layer $i$ is itself interpretable as a probability vector that represents the property of the input $\mathbf{x}$ during the decision process. The previous works that combine tree models with deep neural networks focus on this advantage of hierarchical decisions, as the given examples are clustered into separate bins based on their properties with respect to the target classes. This gives us a novel insight into the given data, separately from the accuracy of predictions.

## 3.4 Theoretical Analysis on Interpretability

It has been taken for granted that tree models are interpretable. However, with the integration of deep neural networks, i.e., representative black-box models, the interpretability of tree models in general becomes questionable. For a better understanding of the interpretability of tree models, we provide a theoretic description of the relationship between the interpretability of decision functions and the whole model with respect to a score function for each feature element in Theorem 1.

**Theorem 1.** *Given layer $l$ of a decision transformer, assume that the transition function $\mathcal{T}_m$ of every layer $m \leq l$ is explainable by a nonnegative score function $s(\cdot)$ such that $s(x_k, v(\mathbf{x}))$ quantifies the contribution of $x_k$ to $v(\mathbf{x})$ as a normalized score, i.e., $\sum_k s(x_k, v(\mathbf{x})) = 1$. Then, the output $\boldsymbol{\pi}_l$ of layer $l$ is explainable by the same score function $s(x_k, \pi_{lj})$ for every $j$.*

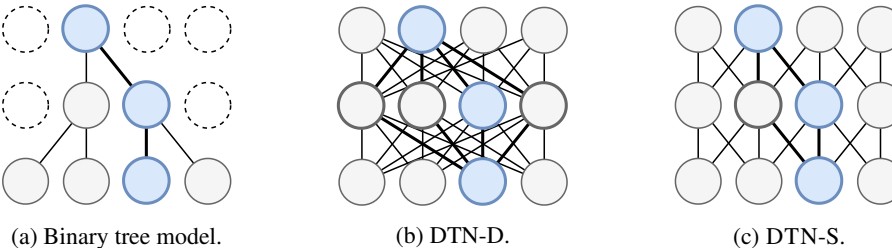

(a) Binary tree model.      (b) DTN-D.      (c) DTN-S.

Figure 2: A comparison between (a) a binary tree model, (b) DTN-D, and (c) DTN-S, assuming three layers each of which has four nodes. A tree model propagates an example through a single path denoted by the bold lines. DTN-D uses all nodes at the internal layer in the propagation, as all pairs of nodes are connected. DTN-S simplifies the structure of DTN-D by imposing a spatial locality, minimizing the number of parameters, still allowing a change of information between nodes.

*Proof.* The following equations hold for positive output values $v_1(\mathbf{x})$ and $v_2(\mathbf{x})$:

$$s(x_k, v_1(\mathbf{x}) + v_2(\mathbf{x})) = v_1(\mathbf{x})s(x_k, v_1(\mathbf{x})) + v_2(\mathbf{x})s(x_k, v_2(\mathbf{x})) \tag{4}$$

$$s(x_k, v_1(\mathbf{x})v_2(\mathbf{x})) = \frac{s(x_k, v_1(\mathbf{x}))s(x_k, v_2(\mathbf{x}))}{\sum_n s(x_n, v_1(\mathbf{x}))s(x_n, v_2(\mathbf{x}))} \tag{5}$$

The first equation is derived from the fact that $x_k$ contributes to the sum of two values based on their sizes and the score for each value. The second equation involves a normalization for all elements of $\mathbf{x}$, as a multiplication does not preserve the normalized scores. Based on these properties, we derive the score $s(x_k, \pi_{li}(\mathbf{x}))$ for the $i$-th element of $\boldsymbol{\pi}_l(\mathbf{x})$ recursively as

$$s(x_k, \pi_{li}) = \frac{1}{z(\mathbf{x})} \sum_j t_{lij}(\mathbf{x}) \pi_{l-1j}(\mathbf{x}) s(x_k, t_{lij}(\mathbf{x})) s(x_k, \pi_{l-1j}(\mathbf{x})), \tag{6}$$

where $t_{lij}(\mathbf{x})$ represents the element at position $(i, j)$ of $\mathcal{T}_l(\mathbf{x})$, and $z(\mathbf{x})$ is a normalization constant. We prove the theorem given that the initial score is a constant as $s(x_k, \pi_{0i}) = 1/|\mathbf{x}|$. $\qquad\square$

**Corollary 3.** *The prediction of a decision transformer $\mathcal{M}$ for any target class $y$ is explainable by a score function $s$ if the transition function $\mathcal{T}_l$ of every layer $l$ is explainable by $s$.*

A decision transformer extends the interpretability of transition functions since they are combined by linear transformations; it is not possible to apply Equation 6 if a nonlinear activation function or a bias term exists between transition matrices. It is notable that our theoretical claim in this section should be distinguished from the structural interpretability of tree models that comes from their property that every hidden layer produces a probability vector. Corollary 3 focuses on how the interpretability of each layer relates to the interpretability of the whole model, based on our understanding of tree models as a decision transformer which aggregates all decisions at each layer as a single operation.

## 4    DECISION TRANSFORMER NETWORKS

Our decision transformer suggests that decision-based models that propagate an input by stochastic decisions are not restricted to a tree structure. We propose Decision Transformer Network (DTN), an extension of tree models into a deep network having a fixed number of nodes at each layer. This solves the limitation of tree models that they have an exponential number of nodes with respect to a tree depth, which prevents them from having a large depth. Our DTN creates multiple paths from a decision node to nodes at the lower layers, improving the representation power, along with its deep structure. Nevertheless, DTN is still interpretable as shown in Corollary 3.

We introduce two versions of DTN. DTN-D is a basic model having full dense connections between layers. DTN-S improves the efficiency and accuracy of DTN-D by utilizing locality-sensitive sparse connections and the spherical softmax function. Figure 2 compares the structures of a tree model, DTN-D, and DTN-S. A tree model creates a single path from a node to another, allowing no change of information between nodes once the branch is split. On the other hand, DTNs generate multiple paths between nodes, improving the representation power of tree models.

Table 2: Tested 121 datasets that are divided into three categories based on the numbers of examples.

| Category | # of Datasets | # of Examples | | | # of Features | # of Labels |
| | | Min | Max | Avg | Avg $\pm$ Std | Avg $\pm$ Std |
|---|---|---|---|---|---|---|
| Large | 9 | 10,992 | 130,064 | 43,943 | 19.0 $\pm$ 15.8 | 8.2 $\pm$ 8.5 |
| Mid | 37 | 1,000 | 8,124 | 3,453 | 40.2 $\pm$ 48.4 | 12.2 $\pm$ 26.6 |
| Small | 75 | 10 | 990 | 371 | 24.4 $\pm$ 37.9 | 4.1 $\pm$ 3.7 |
| All | 121 | 10 | 130,064 | 4,555 | 28.8 $\pm$ 40.8 | 6.9 $\pm$ 15.5 |

## 4.1 DENSE MODEL WITH ALL PAIRWISE CONNECTIONS

Our DTN-D generates a stochastic transition matrix from a weighted linear function of an input $\mathbf{x}$ followed by the softmax activation. We first set $\pi_0 = 1$, making sure that every example is fed to the root node and then distributed to multiple nodes at lower layers. We then define the $(i, j)$-th element $t_{lij}$ of the transition function $\mathcal{T}_l$ at layer $l$ as follows:

$$t_{lij}(\mathbf{x}) = \frac{\exp(\mathbf{w}_{lij}^\top \mathbf{x} + b_{lij})}{\sum_{k=1}^{N_{l+1}} \exp(\mathbf{w}_{lkj}^\top \mathbf{x} + b_{lkj})}, \tag{7}$$

where $\mathbf{w}_{lij}$ and $b_{lij}$ are the learnable parameters for the index $(i, j)$ at layer $l$. We define the classifier function $\mathcal{F}$ also as Equation 7, without separating $\mathcal{T}$ and $\mathcal{F}$ in the network.

## 4.2 SPARSE MODEL WITH LOCALITY-SENSITIVE DECISIONS

We propose DTN-S to improve the efficiency of DTN-D by generating sparse connections between adjacent layers. DTN-S limits the output field of each node to its $W$-hop neighbors at the next layer, imposing a spatial locality between nodes; it makes each node learn its characteristics based on its position. As a result, DTN-S has $O(NLD)$ parameters, which is $N$ times smaller than $O(N^2LD)$ of DTN-D, safely assuming that $W$ is a small constant.[1] $N$ and $D$ represent the number of nodes at each layer and the size of feature vectors, respectively.

At the same time, DTN-S uses the *spherical softmax* function (Martins & Astudillo, 2016) instead of the softmax to support generating sparse probability vectors. The softmax function at Equation 7 cannot produce a zero probability due to the exponential function. On the other hand, the spherical softmax makes each element positive by the square function as $\mathrm{spherical}(\mathbf{z})_i = z_i^2 / \sum_k z_k^2$ instead of the exponential. As a result, along with the locality-sensitive connections, DTN-S can effectively choose the target nodes to pass a given example $\mathbf{x}$ by stochastic decisions.

As a result, each element $t_{lij}$ of the transition function $\mathcal{T}_l$ of DTN-S is defined as follows:

$$t_{lij}(\mathbf{x}) = \frac{(\mathbf{w}_{lij}^\top \mathbf{x} + b_{lij})^2}{\sum_{k=1}^{N_{l+1}} (\mathbf{w}_{lkj}^\top \mathbf{x} + b_{lkj})^2} \text{ if } |i - j| \leq W \text{ otherwise } 0. \tag{8}$$

## 5 EXPERIMENTS

We evaluate our proposed DTN models on tabular datasets where tree models have extensively been adopted. We use 121 datasets of the UCI Machine Learning Repository (Dua & Graff, 2017), which have been used as a benchmark dataset in recent works (Delgado et al., 2014; Olson et al., 2018). We download preprocessed datasets, scale each feature into the zero mean and unit variance, and divide each dataset into the 8:2 ratio for training and test.[2] We categorize the datasets by the number of examples as in Table 1 into *large* ($\geq$ 10K), *medium* ($\geq$ 1K and $<$ 10K), and *small* ($<$ 1K). We run each model five times for each category and report the average and standard deviation of accuracy.

We compare DTNs with existing tree models for tabular datasets, including multilayer perceptrons (MLP) that have been recently tuned for the datasets (Olson et al., 2018) that we experiment with.

---

[1]We fix $W$ to one in all of our experiments to show that even the smallest $W$ performs well.
[2]The datasets are available at http://persoal.citius.usc.es/manuel.fernandez.delgado/papers/jmlr

Table 3: The accuracy of DTNs and competitors on four categories of datasets. The *out of memory* represents the shortage of GPU memory. DTNs show the highest accuracy in most categories.

| Model(-layers) | Large | Medium | Small | All |
|---|---|---|---|---|
| MLP | $91.25 \pm 8.85$ | **$84.51 \pm 13.92$** | $79.08 \pm 17.77$ | $81.65 \pm 16.56$ |
| DT | $84.29 \pm 18.28$ | $80.84 \pm 14.77$ | $77.43 \pm 17.97$ | $78.98 \pm 17.21$ |
| DNDT | $88.63 \pm 6.76$ | $78.30 \pm 17.38$ | $75.23 \pm 16.25$ | $77.16 \pm 16.49$ |
| SDT-10 | $87.23 \pm 14.01$ | $80.78 \pm 16.44$ | $76.88 \pm 17.18$ | $78.85 \pm 17.00$ |
| SDT-12 | $88.10 \pm 13.31$ | $81.69 \pm 15.62$ | $77.25 \pm 17.24$ | $79.41 \pm 16.79$ |
| SDT-14 | $88.63 \pm 12.57$ | $82.11 \pm 15.46$ | $77.49 \pm 17.39$ | $79.73 \pm 16.82$ |
| SDT-16 | | *out of memory* | | |
| **DTN-D (ours)** | $91.13 \pm 7.80$ | $83.09 \pm 14.04$ | $78.19 \pm 17.36$ | $80.65 \pm 16.21$ |
| **DTN-S (ours)** | **$91.40 \pm 7.98$** | $84.22 \pm 13.78$ | **$79.63 \pm 16.87$** | **$81.91 \pm 15.84$** |

MLPs have 2 hidden layers of 128 units, He-initialization (He et al., 2015), ELU activation (Clevert et al., 2016), and dropout (Srivastava et al., 2014).[3] For tree models, we implemented soft decision trees (SDT) (Frosst & Hinton, 2017) and deep neural decision trees (DNDT) (Yang et al., 2018) with PyTorch and used the *scikit-learn* implementation of decision trees (DT). Since DNDTs have an exponential number of parameters with respect to the number of features, we select 20 features for each dataset based on the importance scores computed by DTs. We train the models 200 epochs each using the Adam optimizer (Kingma & Ba, 2015) with the initial learning rates between 0.001 and 0.01: 0.001 in MLPs, 0.01 in SDTs and DNDTs, and 0.005 in DTNs. We ran all experiments on a workstation with GTX 1080 Ti, using PyTorch for GPU computations.

## 5.1 COMPARISON WITH OTHER MODELS

We compare the accuracy of DTNs and competitors in Table 3. The tree models show lower accuracy compared to MLPs, as they focus on the interpretability of decisions rather than the representation power. Even though MLPs maximize the accuracy with a black box structure, our DTNs outperform them in most categories of datasets, while being comparable in the medium-sized datasets. It is also shown from the small standard deviations that the predictions of DTNs are robust as well as accurate, which is an important advantage as an off-the-shelf classifier.

The table also shows the limitation of typical tree models, i.e., a large tree depth cannot be adopted due to an exponential number of nodes. The accuracy of SDTs improves as we increase the number of layers until 14, however, deeper layers after 14 causes the out-of-memory error. As a result, the accuracy of SDTs is worse than that of MLPs. DTs and DNDTs perform worse than SDTs as they use each feature independently by simply comparing its value to a learned threshold. Our proposed DTNs achieve high accuracy while maintaining interpretable as an improved tree model.

## 5.2 INTERPRETABILITY OF DECISIONS

We visualize in Figure 3 the decision maps of DTN-S on the Iris dataset in the small category.[4] The dataset contains four features and three classes for classifying the class of an iris plant. The number of nodes is set to 4 for clear visualization. The figures show that the examples are categorized into random clusters at the root layer, but the clusters become clearer as the number of layers increases, approaching the final classes. The blue and green classes are easily separated by the leaf classifier and thus mapped to the same cluster in Figure 3b. On the other hand, the examples of the orange class have several mixed clusters in Figure 3b, since they are located close to the green class.

---

[3]We have tested deeper layers, however, MLPs showed lower accuracies for layers deeper than 2.
[4]http://archive.ics.uci.edu/ml/datasets/Iris/

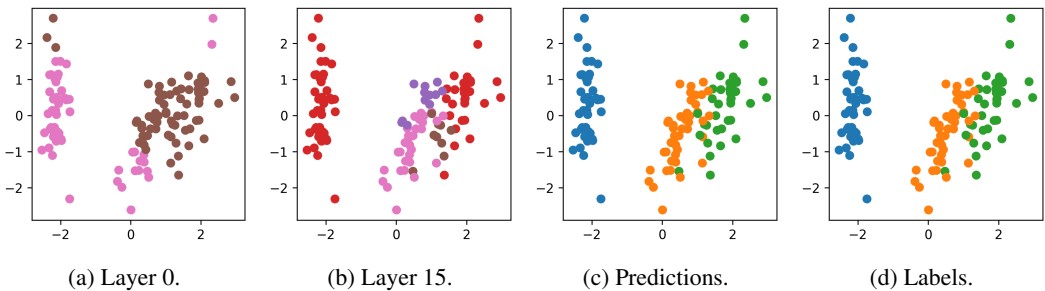

(a) Layer 0.    (b) Layer 15.    (c) Predictions.    (d) Labels.

Figure 3: The decision map at each layer of DTN-S for the Iris dataset in the small category. All feature vectors have been transformed by the principal component analysis (PCA) for visualization. Each point represents a feature vector, whose color represents the node with the maximum membership probability. The decision maps of intermediate layers give valuable insight into the dataset.

Table 4: The accuracy of DTNs with various options of activation and sparsification functions. The spherical softmax consistently outperforms the softmax function, and locality-sensitive connections show similar accuracy to dense connections even with fewer parameters.

| Sparsity(-activation) | Large | Mid | Small | All |
|---|---|---|---|---|
| Dense-softmax (DTN-D) | $91.13 \pm 7.80$ | $83.09 \pm 14.04$ | $78.19 \pm 17.36$ | $80.65 \pm 16.21$ |
| Dense-spherical | $\mathbf{91.72} \pm \mathbf{7.51}$ | $\mathbf{84.56} \pm \mathbf{13.59}$ | $79.07 \pm 16.99$ | $81.69 \pm 15.93$ |
| Pruned-softmax | $90.49 \pm 8.67$ | $83.28 \pm 14.26$ | $79.11 \pm 16.92$ | $81.23 \pm 15.91$ |
| Pruned-spherical | $\underline{91.60} \pm 7.89$ | $83.99 \pm 13.90$ | $\underline{79.44} \pm 16.58$ | $\underline{81.74} \pm 15.61$ |
| Local-softmax | $90.60 \pm 8.20$ | $83.42 \pm 14.19$ | $79.13 \pm 16.81$ | $81.30 \pm 15.89$ |
| **Local-spherical (DTN-S)** | $91.40 \pm 7.98$ | $\underline{84.22} \pm 13.78$ | $\mathbf{79.63} \pm \mathbf{16.87}$ | $\mathbf{81.91} \pm \mathbf{15.84}$ |

## 5.3 ABLATION STUDY OF DTNS

We further compare various activation functions and sparsification methods of DTN in Table 4. We implement an additional baseline of sparse DTN for a fair comparison, which randomly prunes the connections between adjacent layers to maintain the same sparsity as in DTN-S.

The accuracy of DTN improves as we adopt locality-sensitive connections and spherical softmax. The spherical softmax outperforms the softmax in all cases, regardless of the sparsity of connections, showing the importance of generating sparse probability vectors with a simple activation function. The locality-sensitive connections improve the accuracy of dense connections especially in the small datasets, where the efficiency and robustness are important for achieving high accuracy. The random sparse connections show generally lower accuracy than the locality-sensitive connections, due to the random nature that does not consider the positions of nodes in generating probabilities.

## 6 CONCLUSION

In this work, we propose Decision Transformer Network (DTN), an extension of tree models into a deep network, which improves their representation power by generating multiple paths between nodes. In order to devise DTN, we first propose a decision transformer, our novel framework that generalizes existing tree models in the context of deep learning by formulating tree layers as stochastic decisions. Understanding tree models as a decision transformer gives us a theoretical foundation for analyzing the stochastic nature of tree models regardless of the decision functions used.

Two versions of DTN are introduced: DTN-D, a basic dense version, and DTN-S, a sparse version having improved accuracy and efficiency. DTN-D utilizes a series of linear transformations with the softmax activation, while DTN-S adopts locality-sensitive connections and the spherical softmax to improve the accuracy and efficiency of DTN-D. We show that our DTN-S outperforms existing tree models and deep neural networks in 121 tabular datasets, demonstrating its representation power and robustness at the same time while generating interpretable decision processes.

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
