# OpenReview forum: "Generalizing Tree Models for Improving Prediction Accuracy"
_ICLR.cc/2021/Conference — Reject_

### Official Review · AnonReviewer1 · 2020-10-28
**DTNs are not shown to perform better than existing non-interpretable methods**

**Rating:** 4
**Confidence:** 4

**Review:**

Edit:

I have read the authors' response and the other reviews. I still believe that this paper is not ready for acceptance.

Summary:

The authors present "decision transformers," a family of classifiers of which decision trees (DTs) are a subset. They present "Decision Transformer Networks" (DTNs), which are a type of decision transformer where the number of nodes per layer is constant, the decision boundaries are soft, and the nodes are linked more densely than standard DTs. DTNs are experimentally compared to soft DTs, MLPs, and DTs, where the authors believe DTNs perform better than the alternatives. Also, the authors demonstrate that a decision transformer can be explained in terms of a score function if it has no non-linearities or bias terms between transition functions.

Reasons for score:

DTNs are not shown to perform better than existing non-interpretable methods. DTNs are not shown to be interpretable in the sense that tree models are interpretable. DTNs may exhibit a favorable trade-off between accuracy and interpretability, but determining this requires defining and measuring interpretability.

Pros:

-Decision transformers are explained very clearly and may be useful in future analysis of DT-style models.

-Initial results for DTNs are favorable, so DTNs may be further refined and yield a good architecture.

-The ablation study (Section 5.3) is a great inclusion which allows others to extend DTNs.

Cons:

-The authors note that "with the integration of deep neural networks ... the interpretability of tree models in general becomes questionable," but the proposed method is effectively integrating single-layer networks into its many transition matrices. The evaluated models have 16 layers with 128 nodes per layer (obtained from code; not entirely clear from paper text), use soft non-linearities, and have more connections than a tree-structured model. The paper would be improved by a clearer demonstration of the "tree" nature of the proposed model or how it could be interpreted in tree-like ways.

-The theoretical analysis for interpretability does not apply to models "if a nonlinear activation function or a bias term exists between transition matrices". However, the non-linearity and bias terms have simply been moved into the transition matrices. As a result, the score function explanations are oblivious to the contents of the transition matrices.

-Explanations in terms of importance values on input features are also applicable to DNNs; the benefit of a tree model is the structure. Likewise, the benefit of fewer parameters is the ability for a user to simulate a decision or more easily understand the entire model. Neither of these interactions seems possible with a DTN. The authors claim that DTNs preserve the "inherent interpretability of the tree structure," though the tree structure is largely lost. One of the benefits of a hard-boundary tree is that the model's behavior over a subset of the input space can be described by a smaller (shallower) tree. Likewise, a single input traces a single path through the tree, so only one path per layer is relevant. For DTNs, the entire network is applicable to the entire input space. The network cannot be examined for a single region or for a single input without considering the entire network. This downside of DTNs is not evaluated and is largely not addressed.

-The authors aim to create an interpretable method but do not measure how interpretable their method is nor define interpretability. Additional experiments on interpretability would support the authors' claims that their model is interpretable.

-The experimental setup should be improved. The authors cite Olson et al. (2018) for their MLPs, but that work does not consider MLPs with 2 layers of 128 nodes (i.e., the architecture used by the authors of this work). In particular, an MLP with "local" connections (as in DTN-S) with depth 10 and 128 nodes per layer is similar to DTN-S and may perform better than the evaluated MLP. The way hyperparameters are selected should be made more clear to ensure that a fair comparison is made. To demonstrate the high performance of their method, the authors should compare to boosting and/or random forest approaches (not just MLPs).

-Table 3 does not suggest statistically significant improvements over existing methods.

Questions During Rebuttal Period:

Please address and clarify the "Cons" above. In particular: How was the MLP architecture chosen?

Minor Comments:

-In the introduction, the authors note that DTs become wide as the number of layers increases, so deep trees cannot be constructed. However, there exist alternative constructions which limit the width of the tree. For example, all left-children can be constrained to be leaf nodes. This is akin to the limitation of the number of nodes per layer in DTNs.

-Figure 1b has the input on the right and the output on the left. The figure would be clearer if the input was on the left and the output on the right (i.e., flip the figure). Very good figure, though.

-It is clear from context that DTNs have a fixed number of nodes per layer, but this should be made clearer (as this is quite different from standard DTs and the decision transformer shown in Figure 1).

-The width and depth of DTNs is not clearly stated in the paper.

-The ablation study could be moved to the appendix in order to make room for an evaluation of interpretability.

-The paper would benefit from another editing pass for grammar.

Some Typos:

-Section 2: "ensemble methods tboost" -> "ensemble methods boost"

-Section 3.2: "Each leave" -> "Each leaf"

---

> ### Author Response · Authors · 2020-11-23
> **Feedback**
>
> Thank you for the detailed review. We mostly agree with your comments, and thus we’ll answer the question about the structure of MLPs. The structure was chosen by experiments, where we have found that two layers produce the best accuracy on average. Increasing the number of layers of MLPs significantly dropped its accuracy because most datasets in our experiments are quite small compared to typical CV or NLP datasets. The following is the result of our experiments for MLPs (for column All of Table 4):
> - 80.87 \pm 16.52 (L=1)
> - 81.65 \pm 16.56 (L=2)
> - 81.48 \pm 16.90 (L=4)
> - 80.70 \pm 17.30 (L=8)
> - 78.68 \pm 18.78 (L=16)
>
> On the other hand, our DTNs are quite robust to the number of layers, because the transition functions are stacked linearly unlike in deep neural networks.

---

> > ### Comment · AnonReviewer1 · 2020-11-24
> > **Response to MLP Depth Results**
> >
> > Thank you for the additional information! As noted before, adding this to the paper would improve its clarity since Olson et al. (2018) are mentioned but their exact setup is not used.

---

### Official Review · AnonReviewer4 · 2020-10-28
**Poor presentation, contributions are not strong enough**

**Rating:** 4
**Confidence:** 3

**Review:**

The authors propose Decision Transformer Networks (DTNs): a model that generalizes decision trees to deep network style decision graphs. Structurally, DTNs are similar to deep neural networks with layers of nodes operating as stochastic decision functions and which output probabilities of an input vector belonging to each sub-network. The probabilistic predictions of the decision nodes in a layer are aggregated in a "transition matrix" and passed to the next layer until the leaf layer is reached where the final classification decision is made. The goal of DTNs is to improve the representation power of tree models by transforming (exponentially large) trees to compact graphs without losing its interpretability power. The authors propose special scoring functions to explain the decisions made by  each layer as well as the model. On a diverse set of datasets ranging from small (<1K) to large (>10K), DTNs (dense and sparse) with softmax and spherical-softmax decision nodes, attained better accuracies than MLPs, conventional decision trees and their soft versions.

The motivation of the work is quite clear: to improve both the representation power and the interpretability of tree structured decision trees. Unfortunately,  I could not find the contribution(s) significant enough for ICLR.  Decision nodes making probabilistic predictions is not new. There is previous work that the authors cite. The cited works are mostly for tree structured models. But there are graph structured probabilistic models with superior representation capability but may lack interpretability. It seems that DTNs just use the gain in interpretability through the proposed scoring function which seems to be weak  and not generalizable. The paper is lacking in quality and clarity of presentation as well.

The architecture is not well explained. There is no defined root in DTNs, rather a set of first layer nodes which acts as the root layer, yet a single root is shown in Figure 2. Figure 2 is confusing: it is not clear what the blue nodes represent and why only certain edges are shown in bold (b and c). I understood from the description of DTNs that all nodes in a layer participate in making predictions, but the figure shows otherwise. Most of the formal definitions and descriptions in section 3 are focused on decision nodes with only two branches, left and right. But the authors don’t formally and clearly state how these could be generalized to multiple branches when formally introducing DTNs in section 4 and in experiments.

One of the claimed contributions in the paper is the theoretical analysis of interpretability of DTNs (theorem 1). First, the notations used in the theorem are not defined, for example x_k, \pi_{lj}, v_1(x) etc. Next, how does one interpret the score for a layer and for a decision. How does the score help to derive any meaningful explanations for a prediction? There were no experiments done to support this scoring function. The experiment on clustering is something obvious and doesn't show the benefit of using DTNs. Besides it is mentioned that the proposed scoring function does not apply to non-linear activation functions. It seems that this scoring function is weak as it will not work for other models that DTNs can be generalized to.

The question of how to train these models were completely omitted.  The authors only mentioned the activation functions used in nodes. I could not find any details on the architecture or any statistics on the size of these networks.

Other probabilistic models for discriminative tasks:
Sum-product networks: + Gens, Robert, and Pedro Domingos. "Discriminative learning of sum-product networks." Advances in Neural Information Processing Systems. 2012.
+van de Wolfshaar, J. and Pronobis, A., 2019. Deep Generalized Convolutional Sum-Product Networks for Probabilistic Image Representations. arXiv preprint arXiv:1902.06155.
Cutset networks: + Rahman, T., Jin, S. and Gogate, V., Cutset Bayesian Networks: A New Representation for Learning Rao-Blackwellised Graphical Models, in IJCAI 2019.
+Rahman, T., Kothalkar, P., and Gogate, V. "Cutset networks: A simple, tractable, and scalable approach for improving the accuracy of Chow-Liu trees", ECML 2014.

---

> ### Author Response · Authors · 2020-11-23
> **Feedback**
>
> Thank you for the detailed review. (a) The formal definition of DTNs does not contain a root node. Still, we have assumed a single root for simplicity in Figure 2. (b) The blue nodes represent the path of propagation with an assumption that only a single path is activated. (c) We have focused on trees having two children at each node because that is the structure that most tree models adopt. (d) There are several definitions of interpretability of models. We have adopted the notion that a model is interpretable when the importance of each input feature can be measured by the contribution of the feature on the result/decision of the given input. The training was done to minimize the cross-entropy loss of classification problems.

---

### Official Review · AnonReviewer3 · 2020-10-28
**Interesting idea of seing differently decision tree models and how interpret it**

**Rating:** 6
**Confidence:** 4

**Review:**

Summary:
The authors propose a new model that consists in transforming decision trees by seeing such models as a sequence of "layers" made of all nodes at the same tree depth. Each layer is then modelled as a stochastic decision (multiplication by a stochastic matrix) which can mimic the behaviour of both standard decision tree and stochastic trees (among others). Their idea is to keep the layer decomposition as it is to be able to understand/interpret each layer as pieces of the bigger picture (i.e., the interpretation of the model itself).

Overal comment:
The approach is well motivated (comment (a)) and the paper is well-written. The most interesting feature of the proposed approach is the generalisation / unified view of several decision tree algorithms that may appear as completely different and but comes down to a mere choice of a type of decision function (kind of layers) in the proposed approach (similarly to what typically happens when building a neural network). On the other hand, this work is focused on mimicking the deep learning framework while neglecting a bit the tree-based learning framework (comments (b),(c)). Some practical details (comment (f)) or some explanation are however missing for in-depth understanding of the results (comments (c), (d), (e),(f)). Therefore, I think the paper could be accepted for the generalisation of existing approaches, the improvement of prediction accuracy (comment (d)) but the interpretability is somehow limited (comment (e)) despite a strong motivation regarding the interpretability of the proposed approach (comment (a)).

Major comments:
(a) The motivation of focusing on the interpretability of combined tree-based and deep neural networks models is very good as these methods were most likely designed to improve the raw performances without considering the loss of interpretability.
(b) To which extent your method relates with the tree-based learning framework? In particular, you said that "the transition matrices of a decision transformer [...] can be collapsed to form a single transition matrix". I totally agree that for the sake of interpretability you suggest not to consider the model as a single transition matrix but mathematically it can be rewritten as such. Therefore, how does it relates to kernel-based formulation of decision trees models?
(c) It is unclear to me how is defined a "contribution of x_k". There is a large amount of working focusing on the measuring feature contributions from tree-based models. Similarly to (b), how your work relates to this?
(d) Table 3 shows impressive results. However, it would have been interesting (maybe not better but informative at least) on the number of wins/ties/loss of your model (say the best one) wrt to the others. Indeed, given the standard deviations and the mean over many differents problems, it is not clear if your approach is always better or only partially.
(e) I see the point of Figure 3 as showing the evolution of the decision map moving from not well separated samples to easily separable classes. However, the definition of "random clusters" (at the root layer for instance) is not clear as the problem is defined with three classes, I believe that at layer 0 you do not look where samples end and therefore have not idea on the predicted class (hence explaining the "new" colours). It is also surprising to consider "two" clusters at layer 0, then "four colours" at layer 15 while finishing with 3. Please clarify.
(f) It would have been interesting to have more details on the practical details on the learning of your models (time, implementations, ...).

---

> ### Author Response · Authors · 2020-11-23
> **Feedback**
>
> Thank you for the positive review. (b) Our work relates to previous tree models in that a DTN can generalize any tree model by adopting a proper transition function at each layer, even though it uses a complex kernel function at each decision. (c) The contribution of a feature element, i.e., the importance of the feature for a decision, is evaluated by an interpretability score. (d) Following the general observation that there is no single machine learning model that outperforms all others in every data, we observe DTNs do not outperform previous models in some data sets. This just shows that we have tried datasets with various properties and sizes. (e) We set the number of nodes to 4, which means that each layer divides given examples into four clusters, except for the leaf nodes that classify each example into one of the three classes. The root node uses only two clusters while it has four. We can guess that model uses fewer clusters than the actual number of classes due to the low complexity of the dataset.

---

### Official Review · AnonReviewer2 · 2020-10-28

**Rating:** 3
**Confidence:** 4

**Review:**

Update: I have read the author's response and decided to keep my review, confidence, and score.

----

Summary: this paper generalizes existing decision trees to some neural-style model. The most critical argument is that the model generalizes decision tree while maintaining interpretability. Since this is a new model, interpretability should at least be justified with the visualizations that can be judged as interpretable (let alone an objective measurement or human experiments). However, the demonstration of interpretability is far from satisfaction, so I recommend a clear rejection. See below for further details.

1. One of the major motivation is using a linear number of parameters w.r.t. tree depth to construct a decision tree-style model. However, this problem has been investigated in literature but not compared theoretically or empirically at all in this paper. See the classic paper [1] and the recent paper [2].

2. The current proposition 1 does not seem useful. Any classifier $p(y | x)$ can be written as a decision transformer $F \pi_0$ by letting $\pi_0 = [1]$ and $F = p(y | x)$. The factorization only makes intuitive sense when each $T_i$ is restricted / interpretable.

3. Interpreting a stochastically routing decision tree would inevitably involve inspecting the whole tree, since every prediction involves the whole tree. Hence, visualizing the whole tree is necessary to claim interpretability, especially for a new architecture. The only visualization in Fig. 3 is far from visualizing an interpretable model.

4. The theoretical statement is not clear and rigorous. E.g., what do the authors mean by "explainable"?

5. the proposed architectures seems to highly relevant to hierarchical mixture of experts [3], which can be trained via EM algorithms efficiently. Can the authors show similar things here?

[1] Langley, Pat, and Stephanie Sage. "Oblivious decision trees and abstract cases." Working notes of the AAAI-94 workshop on case-based reasoning. 1994.

[2] Lee, Guang-He, and Tommi S. Jaakkola. "Oblique Decision Trees from Derivatives of ReLU Networks." International Conference on Learning Representations. 2019.

[3] Jordan, Michael I., and Robert A. Jacobs. "Hierarchical mixtures of experts and the EM algorithm." Neural computation 6.2 (1994): 181-214.

---

> ### Author Response · Authors · 2020-11-23
> **Feedback**
>
> Thank you for the detailed review. We have assumed a scenario that each input feature T_i is itself interpretable, and thus making a linear stack of decision functions can be thought of as an efficient way of building an interpretable decision. We used the term “explainable” to mean that we can measure the contribution of each feature in the decision, which is a direct way to make each decision interpretable. Our work can be seen as a generalization of the mixture of experts [3], since we have no assumption on the transition or expert functions unlike in [3]. We can formulate an EM algorithm for training our DTNs if we restrict the transition function to a linear function.

---

### Decision · Program_Chairs · 2021-01-07
**Final Decision**

**Decision:**

Reject

**Comment:**

The paper provides a neural generalization of decision trees with the idea of maintaining interpretability. The approach falls a bit short on theoretical grounds. For example, the main theorem portraying interpretability isn't properly defined and some definitions appear implicitly in the proof. The view of decision trees as a sequence of soft decisions appears to need to model how the full probability distribution over the nodes propagates at each depth. A much stronger case for interpretability (rather than assuming that each T_i is interpretable) should be made if this is kept as one of the main arguments for the architecture. Interpretability of decision trees does not directly carry over to these models.